# Smart city pilots, marketization processes, and substantive green innovation: A quasi-natural experiment from China

Zhi Zhang 🔘 *, Chengting Zheng, Longyao Lan

Department of Financial Management, Fuzhou University of International Studies and Trade, Fuzhou, China

* zhangzhi@fzfu.edu.cn

**Data Availability Statement:** All relevant data are within the paper and its Supporting information files.

## Abstract

The world's major economies are striving to control carbon emissions and avoid irreversible impacts on the natural environment. Therefore, innovative green technologies are crucial for both government departments and the private sector as an important way to address carbon emissions. This study aims to investigate the link between the government's smart city construction and corporate green innovation and optimize the policy guidelines that drive green innovation in enterprises. This study analyzes 6,104 panels of Chinese listed companies from 2007–2019. An approach called the Differences-in-Differences model was applied to evaluate hypotheses. The empirical results suggest that smart city pilots drove substantial green innovation in businesses. The marketization process has a moderating effect on the impact of smart city pilots on substantive green innovation in enterprises. Moreover, marketization process has a threshold effect in smart city pilots influencing the substantive green innovation of enterprises, and the effect of smart city drivers influencing the substantive green innovation of enterprises increases significantly when regional marketization process reaches a certain level. The findings of this study provide valuable guidance for policy designers to promote corporate green innovation at both the hardware facility level and the market system level of cities when developing policies related to green innovation.

## 1. Introduction

With the rapid development of the economy, countries are beginning to pay more attention to the environmental problems caused by human activities. Especially in recent years, the tendency toward global warming caused by excessive human development has become a severe test of how to develop high-quality human beings in the future. Human activities are one of the main reasons for the deterioration of the natural environment, which is being accelerated by the excessive consumption of energy and the emission of large amounts of waste. Therefore, it is urgent to change the traditional development model and use resources in a rational and efficient way [1]. In this context, the path of sustainable development is the key to solving the current development problems of human beings. Sustainable development can bring about economic development without causing damage to the environment, and can develop and utilize resources in a reasonable and efficient manner without affecting the future supply of resources for human beings [1–4]. Human activities and behaviors can have a significant

**Funding:** Funding:Initials of the authors who received each award: ZZ. Grant numbers awarded to each author: FJ2021B169. The full name of each funder: Social Science Fund of Fujian, China. URL of each funder website: https://www.fjskl.org.cn/ The funders had no role in study design, data collection and analysis, decision to publish, or preparation of the manuscript.

**Competing interests:** The authors have declared that no competing interests exist.

impact on sustainable development [4–8]. Cities are places where people gather, work, and live and have a vital impact on natural ecosystems. In 2012, China's Ministry of Housing and Urban-Rural Development (MOHURD) issued the "Interim Management Measures for National Smart Cities Pilot". MOHURD announced the first list of smart cities, containing 90 cities, in January 2013. Nearly 10 years have passed since the first batch of smart cities was piloted, and their construction has had significant positive impacts on regional economic development and environmental protection [9, 10].

Researchers argued that smart city construction can enhance production efficiency through technological innovation, thus promoting urban economic growth [11, 12]. Regarding environmental protection, for example, Chu et al. (2021) found that smart cities have advantages in resource allocation and resource efficiency and can significantly reduce urban pollution emissions [10]. When it comes to sustainability, the researchers believe the sustainability features of smart cities are worth exploring. Yigitcanlar et al. (2019) conclude from a comprehensive analysis of the currently available literature on smart cities that smart cities are a technologically cutting-edge, practically complex, and conceptually temporary urban construction program that does not reflect the overall goal of sustainability [13]. In the opinion of D'Auria and Anna (2018), based on the current understanding of the smart city concept, it is still not possible to judge whether smart cities have sustainability features [14]. Although urban development sustainability is not one of the driving forces behind smart city construction, the smart city framework provides an opportunity for sustainable urban development because smart cities are closely related to technological innovation [15]. Beyond the issue of urban sustainability, scholars have conducted extensive research on the impact of smart city drivers on urban development. Existing research indicates that smart city drivers have a significant positive impact on development quality, innovation capability, and green city transformation [16–18]. While most of these studies stay at the city level, a few extend the impact of city building to the firm level.

It has been controversial whether smart city construction can lead to sustainable urban development. Smart cities are the solution to growing social and environmental problems through efficient digital infrastructure. Based on the concept and efficacy of smart cities, they are rapidly becoming a major future development direction for major cities around the world [19]. Scholars in the field of urban research question, from the perspectives of environmental protection and social equity, whether digital technologies led by smart city construction can truly bring about sustainable urban development [20–22]. There is a perception that smart cities are obsessed with the application of new technologies and focus on rapid economic development rather than environmental sustainability. Therefore, the effectiveness of smart city construction and sustainable development goals are inextricably linked. Misconceptions about smart city construction goals can lead to irrational investments of limited funds in cities [15]. Smart city skeptics argue that solving urban problems has nothing to do with technology. Human thinking, managerial policies, and planning all affect the sustainability of cities [23, 24]. The construction goals set by the government can influence the green innovation behavior of companies [25]. Green innovation in business is an important factor influencing sustainable urban development [26]. In addition, the innovative behavior of firms is usually regulated by the market; for example, market competition and market resource allocation can influence firms' motivation to innovate [27]. Thus, in a country where the market-based reform process has made the market play a decisive role in resource allocation, corporate green innovation may be affected accordingly. Therefore, based on the above considerations, we use marketization as a key variable to test whether and how marketization moderates the construction of smart cities and impacts green innovation. Therefore, this study aims to investigate the link between the government's smart city construction, the degree of regional marketization, and corporate green innovation.

Although previous papers have given considerable attention to the environmental effects of smart city construction, few studies have explored whether smart city construction is associated with the green innovation behavior of firms. This is detrimental to our understanding of the environmental protection effects of smart city construction, especially in developing countries that face severe pollution. This paper attempts to fill this gap. The marginal contributions of this paper include two main aspects. (1) This paper explores the effects of smart city construction on corporate green innovation and describes the corresponding mechanisms by which smart city construction affects corporate green innovation. This adds value to the theoretical and empirical exploration of the environmental protection effects of smart city construction. (2) In this paper, the moderating role and threshold effect of the degree of marketization are explored when studying the impact of smart city construction on corporate green innovation. This is both a response to the question of whether market-oriented reforms in China are conducive to sustainable development and a theoretical supplement to previous literature, broadening the scope of research on the factors influencing corporate green innovation.

The remainder of this paper is organized as follows: Section 2 is a literature review and hypothesis development. Section 3 presents the research methodology, data, and variable definitions. Section 4 contains the empirical results. Section 5 is the discussion. Section 6 is about the conclusions. Section 7 outlines the policy implications. Section 8 is about the limitations and future research.

## 2. Literature review and hypothesis development

### 2.1 Smart cities and substantive green innovation in enterprises

Smart city construction as a government action will have an impact on enterprises in three dimensions: technology, human capital, and institutions [28]. Smart cities are mainly used to promote smarter and more efficient infrastructure through the use of modern communication and Internet of Things technologies to provide more efficient research facilities for research institutions and enterprises [29]. Efficient smart city systems and more advanced intelligent innovation systems are more attractive to innovative talents. Moreover, in the process of building a smart city, the government needs to continuously attract human resources through financial subsidies and policy incentives to create a talent gathering effect, which will facilitate the gathering of innovative talents in enterprises and give better play to the city's innovation function [18]. Finally, smart city construction will lead to more responsive and intelligent environmental regulation. Unlike the traditional urban air quality monitoring system based on ground-based monitoring and satellite data, smart cities will introduce a new generation of information technology [30], and more sensitive and strict environmental regulatory measures will force companies to reduce carbon emissions and implement green innovation strategies. Therefore, we present Hypothesis 1.

Hypothesis 1: Smart city pilots drive substantive green innovation in enterprises.

### 2.2 The role of marketization in the impact of smart cities on green innovation in enterprises

The theoretical relationship between the market as the "invisible hand" and the government as the "visible hand" can be traced back to the economic theories of Keynes and Adam Smith. Keynes considered Adam Smith's model of a market without government involvement to be a special case, an ideal state. Keynes believed that government intervention could play an important role when the market mechanism failed; for example, in times of economic depression, loose

government monetary and fiscal policies could activate market dynamics and promote economic growth. This also implies that government intervention and market mechanisms can work together to produce good results for economic development. The marketization process is then introduced for analysis as a moderating effect of the smart city pilots influencing enterprises substantive green innovation. The marketization process refers to the level and degree of marketization in a country or region [31]. Marketization specifically refers to the allocation of market resources according to market rules, where factors of production (capital, labor, and land) and products are freely acquired and traded by the market without direct or lesser government intervention. From the 1970s on, China began its market-oriented transformation from a centrally planned economy to a market-based economy. China has since gradually transformed from a non-market-oriented system to a market-oriented one, which also means that China has made major adjustments in the way it allocates resources, in macro-economic regulation and government control, and in the structure of its economy [32]. There are, however, large differences in the level of marketization between regions in China, with different degrees of local protection, free markets, and government intervention in different regions, and a comprehensive assessment of the marketization process may reflect the level that is being achieved in the region [33]. According to previous studies, the regional marketization process affects the flow of innovation resources and contract signing in smart cities, and a higher level of marketization implies a higher degree of freedom for enterprises, less government intervention, and the ability of enterprises to boost vitality through free market competition, which can affect the effect of smart city pilots on enterprises' roles in substantive green innovation [32, 34, 35].

Institutional theory indicates that imperfect market-based mechanisms often tend to lead to weak awareness of property rights protection, an opaque institutional environment, and corruption, which ultimately have direct negative impacts on contract signing and market transaction costs. In other words, as market participants, enterprises' micro-decisions are inevitably influenced by the market environment [27]. Therefore, when the degree of marketization is low, enterprises can survive without actively adapting to changes in the smart environment, a phenomenon that restricts the smart city environment's ability to improve the level of green innovation in enterprises. In the unfair market system, the competition mechanism is not available, and the vitality of enterprises is lost, which limits the effect of smart city pilots on the impact of substantive green innovative enterprises. However, when the regional marketization level is high, the market environment is fair, the legal market system is relatively perfect, the real estate and intellectual property rights of enterprises can be effectively protected, the transaction costs of enterprises are low, and it is convenient for enterprises to sign contracts and fulfill contracts, the perfect market mechanism has a positive effect on the impact of smart city pilots on the substantive green innovation of enterprises [27, 32, 35, 36]. Thus, based on the above analysis, this paper proposes the following hypotheses:

Hypothesis 2: The marketization process has a moderating effect in the smart city pilots, influencing enterprises' substantive green innovation.

Hypothesis 3: The moderating effect of the marketization process in smart city pilots affects the substantive green innovation of enterprises and has a threshold characteristic.

## 3. Methodology, data, and variable definition

### 3.1 Methodology

Using the Chinese smart city pilots as a quasi-natural experiment, we apply the propensity score matching method and the difference-in-differences method to measure the impact of

smart city drivers on the background green innovation of enterprises. This paper used listed companies with office addresses in pilot cities as the experimental group and listed companies with office addresses in non-pilot cities as the control group. The propensity score matching method proposed by Rosenbaum and Rubin (1983) [37] is used to match samples to avoid selective sample bias, thereby improving the reliability of the regression results of the difference-in-differences model. During the modeling process, two types of virtual variables were designed: (1) Treatment and control groups; we take enterprises in smart city pilot cities as 1 and other enterprises as 0. (2) Time virtual variable; we take 2013 and after as 1 and before as 0, because China announced the smart cities pilot list in January 2013.

While enterprises' green patents are often used as a proxy variable for enterprises' substantive green innovation, green patent types mainly include green invention patents and green utility model patents, and in general, enterprises' patents with green invention have larger and more substantive effects on them. Therefore, we follow the literature [38–40] to measure enterprises' substantive innovation by the number of green invention patents and their non-substantive innovation by the number of patents in the green utility model, and we construct Model 1 to estimate the impact of the smart city pilot on enterprises' substantive green innovation.

$$Lngreen\_patent_{it} = \beta_0 + \beta_1 Smart_{it} \times Post_{it} + Control_{it} + \xi_i + \lambda_t + \varepsilon_{it} \tag{1}$$

Where $Lngreen\_patent_{it}$ is the natural logarithm of the number of corporate green invention patent applications plus 1, $Smart_{it}$ is the smart city dummy, $Post_{it}$ is the pilot time dummy, and $Control_{it}$ is the control variables that included firm-level as well as city-level variables, as shown in Table 1. We further consider firm fixed effects and year fixed effects. Standard errors are clustered at the city level. We winsorize continuous variables at the 1st and 99th percentiles.

**Table 1. Variable names and definitions.**

| Variable type | Variable name | Variable symbol | Variable description |
|---|---|---|---|
| Dependent variables | Number of green invention patent applications by enterprises | Lngreen_patent | Natural logarithm of the number of green invention patents applied for by enterprises plus 1 |
| | Number of green utility model patent applications by enterprises | Lngreei_upatent | Natural logarithm of the number of green utility model patents applied for by enterprises plus 1 |
| Independent variable | The interaction term between smart city pilot and policy time. | Smart×Post | **Smart** is the pilot city dummy variable, which takes 1 when the city where the enterprise is located is the pilot city, otherwise it takes 0. **Post** is the policy implementation time dummy variable, which takes 1 in 2013 and after, otherwise it takes 0. |
| Control variables | Profitability | Roa | Net profit divided by total assets |
| | Debt level | Lev | Total liabilities divided by total assets |
| | Enterprise size | Lnsize | Natural logarithm of total assets |
| | Cash flow | Cash | Net cash flow from operating activities divided by total operating revenue |
| | Growth | Gro | Tobin's q. |
| | Concentration of shareholding | H1 | Shareholding ratio of the largest shareholder |
| | Percentage of independent directors | Inde | Number of independent directors divided by total number of board of directors |
| | GDP per capita | Lnpgdp | Natural logarithm of annual GDP per capita |
| | Share of tertiary sector | Third | Annual tertiary sector value added divided by GDP |
| | Urbanization | Urb | Year-end urban population divided by year-end entire urban population |
| | City size | Lncityp | Natural logarithm of urban population at the end of the year |
| | Level of financial development | Fdel | RMB loan balance of financial institutions at the end of the year divided by GDP |
| | Degree of urban openness | Open | Annual actual utilization of foreign capital divided by GDP |
| | Level of informationization of the city | Lnfl | Number of Internet access users divided by year-end urban population |

## 3.2 Data

In our initial data, we included enterprises listed on the Shanghai and Shenzhen exchanges. Given the implementation of new corporate accounting standards in China from January 2007, the period 2007–2019 was chosen as the sample period to ensure the comparability of the corporate financial data. The relevant financial data of the enterprises is obtained from their public financial statements. The data on enterprises' green inventions are obtained from the China National Intellectual Property Administration (CNIPA) and the World Intellectual Property Organization (WIPO). The following methods were used to exclude: (1) enterprises that have been listed for less than one year; (2) enterprises with incomplete data on key variables; and (3) enterprises from banking, securities, insurance, futures, and other financial industries. By following the above steps, we obtained a total of 6,104 firm-year observations.

## 3.3 Variable definition

The dependent variable is the firm's substantive green innovation. Following the substantive green innovation literature [38–40], we measured substantive green innovation of enterprises using green invention patents (*Lngreen_patent*) and measured non-substantive green innovation of enterprises using green utility model patents (*Lngreen_upatent*). The independent variable is smart city pilot, and a dummy variable measure is constructed.

In terms of control variables, following the substantive green innovation literature [38–40], we control a series of variables that may affect the substantive green innovation of enterprises, including return on assets (*Roa*), liability ratio (*Lev*), firm size (*Lnsize*), firm cash flow level (*Cash*), firm growth (*Tobinq*), equity concentration (H1), proportion of independent directors (*Inde*), level of urban development (*Lnpgdp*), urban industrial structure (*Third*), degree of urbanization (*Urb*), city size (*Lncityp*), level of financial development of the city (*Fdel*), city openness degree (*Open*), and the level of urban informatization (*Lnfl*).

## 4. Empirical results

### 4.1 Descriptive statistics

The descriptive statistics are presented in Table 2. The variable Lngreen_patent has a sample mean of 0.413, a standard deviation of 0.798, a minimum value of 0, and a maximum value of 6.894, indicating that different enterprises have large differences in green invention patents. In terms of *Lngreen_upatent*, the mean value is greater than that of *Lngreen_patent*, which indicated that enterprises preferred green utility model patents.

**Table 2. Descriptive statistics.**

| Variable | Mean | S.D. | Minimum | Maximum | Observations |
|---|---|---|---|---|---|
| Lngreen_patent | 0.413 | 0.798 | 0.000 | 6.894 | 6 104 |
| Lngreen_upatent | 0.469 | 0.834 | 0.000 | 6.442 | 6 104 |
| Roa | 0.036 | 0.061 | -0.247 | 0.244 | 6 104 |
| Lev | 0.415 | 0.210 | 0.046 | 0.887 | 6 104 |
| Lnsize | 21.896 | 1.198 | 19.548 | 25.309 | 6 104 |
| Cash | 0.084 | 0.185 | -0.753 | 0.677 | 6 104 |
| TobinQ | 2.027 | 1.287 | 0.887 | 8.928 | 6 104 |
| H1 | 0.339 | 0.145 | 0.094 | 0.728 | 6 104 |
| Inde | 0.378 | 0.067 | 0.250 | 0.600 | 6 104 |
| Lnpgdp | 10.963 | 0.577 | 8.703 | 12.165 | 6 104 |

## 4.2 Main results

Table 3 reports the results of the baseline regression. The dependent variables in columns (1)-(3) are green invention patents. The regression results in columns (1)-(3) show that the regression coefficients of *Smart×Post* are significantly positive, indicating that the smart city pilot policy has a positive effect on the substantive green innovation of enterprises. The dependent variables in columns (4)-(6) are green utility model patents. The regression results in columns (4)-(6) show that the regression coefficient of *Smart×Post* is positive, but not significant. From the above results, it can be concluded that the Smart City Pilot has a significant positive effect on the substantive green innovation of enterprises and that the promotion effect on the non-substantive green innovation of enterprises is insignificant. Therefore, hypothesis 1 is verified.

## 4.3 Robustness tests

**4.3.1 PSM-DID test.**   To achieve more robust results, the PSM method and the difference-in-differences method were used to measure the impact of the smart city pilot on enterprises' substantive green innovation.

Figs 1 and 2 show the trend graphs of enterprises' green invention patents and green utility model patents, respectively, where the horizontal axis is the year and the vertical axis is the mean value of green invention patents and green utility model patents, respectively. It can be seen from Fig 1 that the difference between the green invention patents of enterprises in the treatment group and those in the control group was not significant between 2007 and 2010

**Table 3. Baseline results.**

|  | Lngreen_patent | | | Lngreen_upatent | | |
|---|---|---|---|---|---|---|
|  | (1) | (2) | (3) | (4) | (5) | (6) |
| Smart×Post | 0.098** (0.043) | 0.087** (0.045) | 0.091** (0.032) | 0.052 (0.279) | 0.0416 (0.334) | 0.0487 (0.217) |
| Roa |  | 0.275 (0.198) | 0.288 (0.187) |  | 0.124 (0.630) | 0.132 (0.615) |
| Lev |  | -0.170* (0.097) | -0.174* (0.092) |  | -0.0944 (0.460) | -0.0996 (0.444) |
| Lnsize |  | 0.240*** (0.000) | 0.244*** (0.000) |  | 0.239*** (0.000) | 0.242*** (0.000) |
| Cash |  | -0.0181 (0.724) | -0.021 (0.680) |  | -0.0703 (0.249) | -0.0701 (0.258) |
| Tobinq |  | -0.000433 (0.975) | 0.001 (0.947) |  | 0.0107 (0.296) | 0.0119 (0.251) |
| H1 |  | -0.130 (0.551) | -0.126 (0.558) |  | 0.0390 (0.877) | 0.0429 (0.866) |
| Inde |  | 0.253** (0.041) | 0.250** (0.046) |  | 0.0871 (0.591) | 0.0844 (0.606) |
| Lnpgdp |  |  | -0.032 (0.708) |  |  | -0.174* (0.093) |
| Third |  |  | 0.003 (0.413) |  |  | -0.00386 (0.227) |
| Urb |  |  | -0.087 (0.638) |  |  | -0.0189 (0.888) |
| Lncityp |  |  | 0.130 (0.323) |  |  | -0.0137 (0.921) |
| Fdel |  |  | -0.024 (0.399) |  |  | -0.0141 (0.633) |
| Open |  |  | 2.888 (0.697) |  |  | 5.543 (0.477) |
| Infl |  |  | -0.0001 (0.969) |  |  | -0.000536 (0.691) |
| Constant | 0.391*** (0.000) | -4.580*** (0.000) | -5.437*** (0.002) | 0.457*** (0.000) | -4.811*** (0.000) | -2.691 (0.177) |
| Year FE | Yes | Yes | Yes | Yes | Yes | Yes |
| Firm FE | Yes | Yes | Yes | Yes | Yes | Yes |
| Observations | 6 097 | 6 097 | 6 097 | 6 097 | 6 097 | 6 097 |
| Adj. $R^2$ | 0.593 | 0.607 | 0.608 | 0.578 | 0.590 | 0.591 |

Note:

***, **, * indicate significant at 1%, 5%, 10% confidence level, and P values in parentheses.

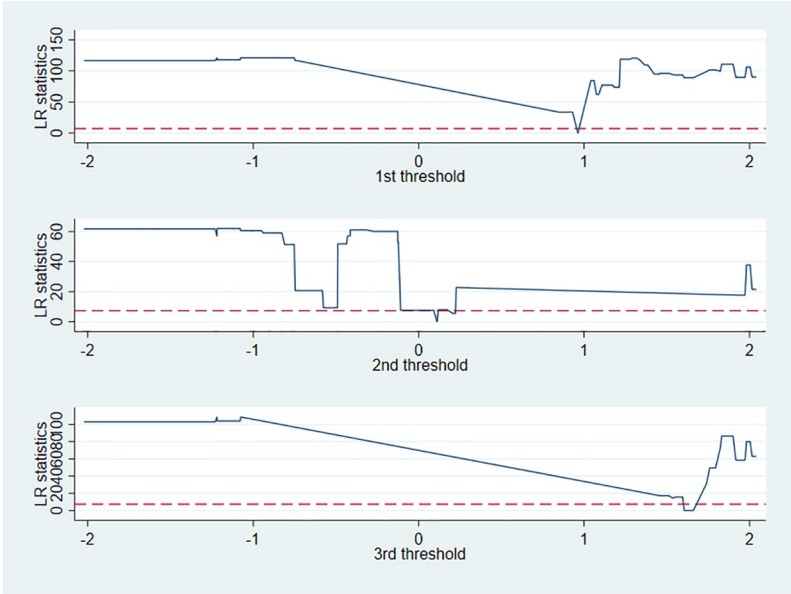

**Fig 1. The trend of green invention patents.**

before the implementation of the policy of the smart city pilot, but the difference between the green invention patents of enterprises in the treatment group and those in the control group was stable between 2011 and 2013. After 2013, the difference between the treatment group and the control group showed a large change, and the policy of the smart city pilot effect was obvious. From the trend of green utility model patents in Fig 2, the difference between the

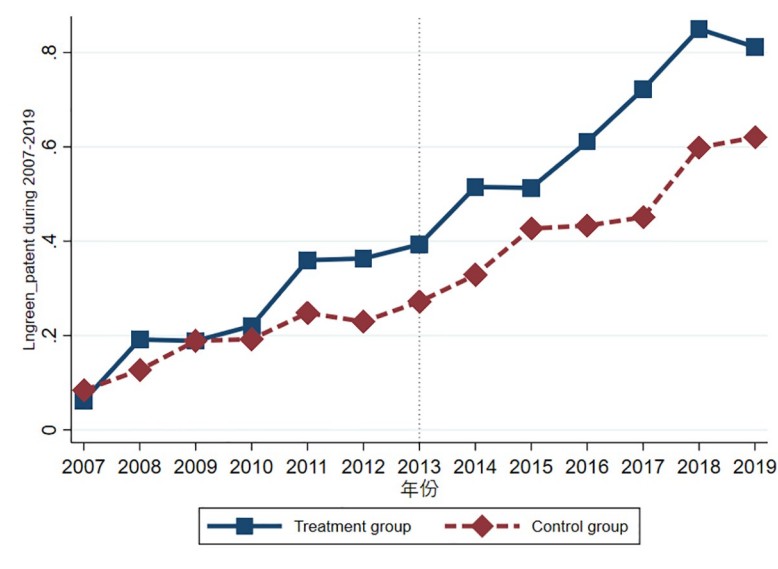

**Fig 2. The trend of green utility model patents.**

treatment and control groups in green utility model patents shrinks rapidly after the policy shock, which is just the opposite of the policy shock effect in Fig 1.

Table 4 shows the difference-in-differences results after PSM. Results show that the Smart×Post coefficients and significance of both green invention patents and green utility model patents do not change substantially compared with the baseline regression, and it remains that the smart city pilot policy has only a positive and significant effect on the green invention patents of enterprises, indicating that the smart city pilot policy does indeed help to promote green innovation among enterprises in the pilot city.

**4.3.2 Instrumental variable method.** To further address the endogeneity issue, we use the instrumental variables (IV) approach for robustness testing. Because information technology is the foundation of smart city construction, we chose the number of international Internet accesses as the instrumental variable for smart cities. The greater the number of international Internet accesses, the higher the level of informationalization, and the more likely it is to be included in the smart city pilot. The number of international Internet accesses we chose in 2001 was not directly related to corporate green innovation in 2019, so we used the number of international Internet accesses in 2001 for each prefecture-level city as an instrumental variable for smart cities and used two-stage least squares for estimation.

Table 5 shows the estimation results for the instrumental variables. The regression estimation results of the first stage show that the regression coefficient of IV is 0.152 and significant. The regression results of the second stage show that the regression coefficient of Smart×Post is 0.321 and significant. The regression results of the first stage show that the F-value is 688.83, which is much greater than 10, indicating that the instrumental variable is not a weak instrumental variable. The regression results in Table 5 under the instrumental variables method indicate that smart city construction has a significant contribution to corporate green innovation, which is consistent with the findings of the baseline regression.

**4.3.3 Placebo test.** To reflect the pilot's policy effect more robustly, we adopted a placebo test. More precisely, the t-value of the policy effect obtained by randomly selecting cities as smart city pilots and cycling them 1,000 times in this manner is presented in Fig 3. Fig 3 shows that the t-value is clustered around 0, while the t value of the policy effect of pilot cities in the real setting is 2.180, meaning that its coefficient is not significant at the 95% confidence level when randomly assigned cities as pilot cities. It thus shows that the effect of promoting substantive green innovation in enterprises exists only in the real-world context of the smart cities pilot, and that this effect is not coincidental.

## 4.4 Underlying plausible mechanism

As described in Section 2.2, we assume that the smart city pilot will promote substantial innovation behavior in firms through two channels: government innovation subsidies to firms and

**Table 4. Regression results of PSM-DID.**

|  | Lngreen_patent | Lngreen_upatent |
| --- | --- | --- |
|  | (1) | (2) |
| Smart×Post | 0.0971** (0.026) | 0.0550 (0.141) |
| Control Variables | Yes | Yes |
| Constant | -6.071*** (0.001) | -4.240** (0.036) |
| Year FE | Yes | Yes |
| Firm FE | Yes | Yes |
| Adj. $R^2$ | 0.608 | 0.587 |

**Table 5. Regression results of IV.**

|  | First-stage Regression | Second-stage Regression |
|---|---|---|
|  | Smart×Post | Lngreen_patent |
| IV: Natural logarithm of Internet access in 2001 | 0.152*** (26.25) |  |
| Smart×Post |  | 0.321*** (3.52) |
| Control Variables | Yes | Yes |
| Year FE | Yes | Yes |
| Firm FE | Yes | Yes |
| F Value | 688.83 | 14.83 |

digital transformation of firms. Using mediating effects models (2), we test whether the smart city pilot has the two channels mentioned above to influence firms' innovation.

$$Subsidies\,(Digitization)_{it} = \beta_0 + \beta_1 Smart_{it} \times Post_{it} + Control_{it} + \xi_i + \lambda_t + \varepsilon_{it}$$

$$Lngreen\_patent_{it} = \beta_0 + \beta_1 Smart_{it} \times Post_{it} + \beta_2 Subsidies(Digitization)_{it} + Control_{it} + \xi_i + \lambda_t + \varepsilon_{it} \qquad (2)$$

Table 6 reports the results of examining the underlying channels. The dependent variables in column (1) are the innovation subsidies given to firms by the government. The regression results in column (1) show that the regression coefficients of Smart×Post are significantly positive, indicating that the smart city pilot has significantly increased the government's efforts to give innovation subsidies to companies. The dependent variables in columns (2) are the degree of digital transformation of the firm. The regression results in columns (2) show that the regression coefficient of Smart×Post is positive and significant, indicating that the smart city pilot has significantly promoted the digital transformation of enterprises. The dependent variables in columns (3)-(4) are green invention patents. The regression results in columns (3)-(4) show that the regression coefficients for subsidies and digitization are significantly positive,

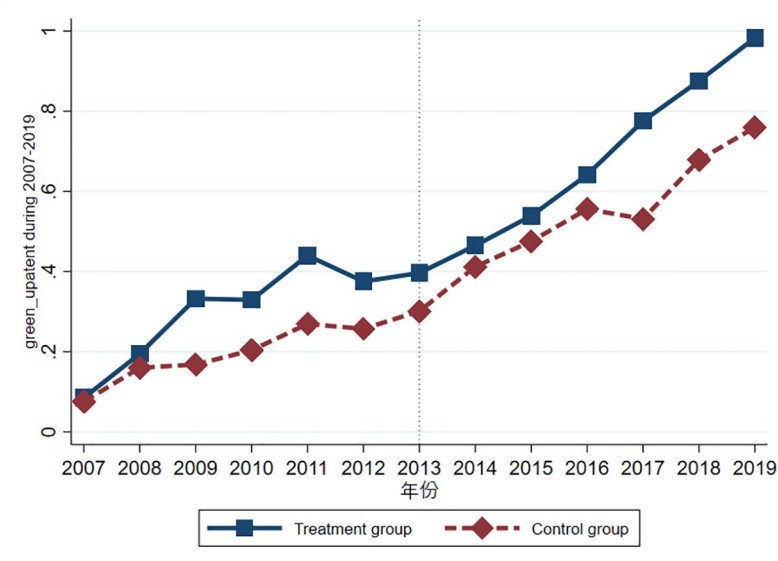

**Fig 3. The result of placebo test.**

**Table 6. Verifying the underlying channels.**

|  | Subsidies | Digitization | Lngreen_patent | |
|---|---|---|---|---|
|  | (1) | (2) | (4) | (5) |
| Smart×Post | 0.460*** (0.000) | 2.477*** (0.000) | 0.086*** (0.007) | 0.070** (0.022) |
| Subsidies |  |  | 0.009** (0.018) |  |
| Digitization |  |  |  | 0.005*** (0.000) |
| Control variables | Yes | Yes | Yes | Yes |
| Constant | 0.788 (0.839) | -38.044** (0.012) | -6.552*** (0.000) | -5.372*** (0.000) |
| Year FE | Yes | Yes | Yes | Yes |
| Firm FE | Yes | Yes | Yes | Yes |
| Adj. R$^2$ | 0.394 | 0.563 | 0.607 | 0.610 |

Note:

\*\*\*, \*\*, \* indicate significant at 1%, 5%, 10% confidence level, and P values in parentheses.

indicating that government subsidies and the digital transformation of enterprises are two important channels through which smart city pilots promote green innovation in enterprises.

## 4.5 Moderating effect test

As a test of the moderating effect of the regional marketization process on the impact of the smart city pilot on substantive green innovation, Model 3 was designed for estimation.

$$Lngreen\_patent_{it} = \beta_0 + \beta_1 Smart_{it} \times Post_{it} + \beta_2 Market_{it} + \beta_3 Smart_{it} \times Post_{it} \times Market_{it}$$
$$+ Control_{it} + \xi_i + \lambda_t + \varepsilon_{it}$$

(3)

$Market_{it}$ is the marketization degree of the city where the firm is located. The marketization degree index is calculated by referring to Fan et al. (2003). Table 7 reports the regression results of the moderating effects model.

The dependent variables in columns (1) and (2) of Table 7 are green invention patents and green utility model patents, respectively. Derivation of $Smart_{it} \times Post_{it}$ inside model 3 yields

**Table 7. Regression results of moderating effects model.**

|  | Lngreen_patent | Lngreen_upatent |
|---|---|---|
|  | (1) | (2) |
| Smart×Post | 0.0391 (0.422) | 0.0108 (0.836) |
| Market | 0.0595 (0.137) | 0.0240 (0.492) |
| Smart×Post×Market | 0.0315* (0.059) | 0.0240 (0.104) |
| Control Variables | Yes | Yes |
| Constant | -5.3296*** (0.004) | -2.5424 (0.216) |
| Year FE | Yes | Yes |
| Firm FE | Yes | Yes |
| Adj. R$^2$ | 0.609 | 0.644 |

Note:

\*\*\*, \*\*, \* indicate significant at 1%, 5%, 10% confidence level, and P values in parentheses.

the following equation:

$$\frac{\partial Lngreen\_patent_{it}}{\partial(Smart_{it} \times Post_{it})} = \beta_1 + \beta_3 \, Market_{it} = 0.0391 + 0.0315 Market_{it} \tag{4}$$

This equation implies that when $Market_{it}$ equals 0, the marginal effect of the smart city pilot on substantive green innovation is 0.0391, and the p-value of the regression result indicates that it is not statistically significant. In fact, the marketization degree is generally greater than 0, so the economic significance of 0.0391 is not significant. $\beta_1$(0.0391) insignificant indicates that when the degree of marketization is low, the impact of conducting a smart city pilot in that city on the innovation of firms is not significant.

The derivative of $Market_{it}$ in Eq (4) gets the following equation:

$$\frac{\partial\left(\frac{\partial Lngreen\_patent_{it}}{\partial(Smart_{it} \times Post_{it})}\right)}{\partial Market_{it}} = 0.0315 \tag{5}$$

A positive and significant coefficient of 0.0315 in Eq (5) indicates that the impact of smart city pilots on substantive green innovation is strengthened when the marketization level is increased. In column (1), the regression coefficient of the $Smart \times Post \times Market$ is significantly positive, indicating that the higher the degree of marketization, the more important the role of the smart city pilot is in promoting substantive green innovation. And the regression coefficient of the $Smart \times Post \times Market$ in column (2) is not significant, indicating that there is no such effect on enterprises with non-substantive green innovativeness. Hypothesis 2 is thus verified.

## 4.6 Threshold effect test

To test the moderating effect of the regional marketization process on the impact of the smart city pilot on substantive green innovation, we developed Eq (6) for estimation.

$$Lngreen\_patent_{it} = \beta_0 + \beta_1 Smart_{it} \times Post_{it} \times I(Market_{it} \leq \gamma_1) + \beta_2 Smart_{it}$$
$$\times Smart_{it} \times Post_{it} \times I(Market_{it} > \gamma_1) + Control_{it} + \xi_i + \lambda_t + \varepsilon_{it} \tag{6}$$

Where $I(\cdot)$ is an indicator function, and $\gamma$ is the threshold value to be estimated. Eq (6) is a single threshold model, from which a multi-threshold model can be extended.

Table 8 reports the test results of the threshold effect model (Eq 6). Single, double, and triple thresholds were tested using $Market_{it}$ as the threshold variable. As shown in Table 8, only the single threshold test results are significant, corresponding to a threshold, F value, and P value of 0.9629, 213.45 and 0.02, respectively, but the double and triple thresholds fail to pass the criterion. As can also be observed in Fig 4, only the single threshold passed the test.

Table 9 reports the regression results of the threshold effect model. The regression coefficient of $Smart \times Post$ is 0.014 when the value of $Market_{it}$ is smaller than the single threshold. The $Smart \times Post$ regression coefficient is 0.123 when the marketization level is greater than

**Table 8. The test results of threshold effect.**

| Threshold type | Threshold value | F-statistics | P-value | Critical value (10%) | Critical value (5%) | Critical value (1%) |
|---|---|---|---|---|---|---|
| Single-threshold | 0.9629 | 213.45** | 0.02 | 150.334 | 160.101 | 326.608 |
| Double-threshold | 0.1112 | 55.99 | 0.30 | 78.299 | 86.437 | 117.605 |
| Triple-threshold | 1.6055 | 97.45 | 0.44 | 206.216 | 267.496 | 420.704 |

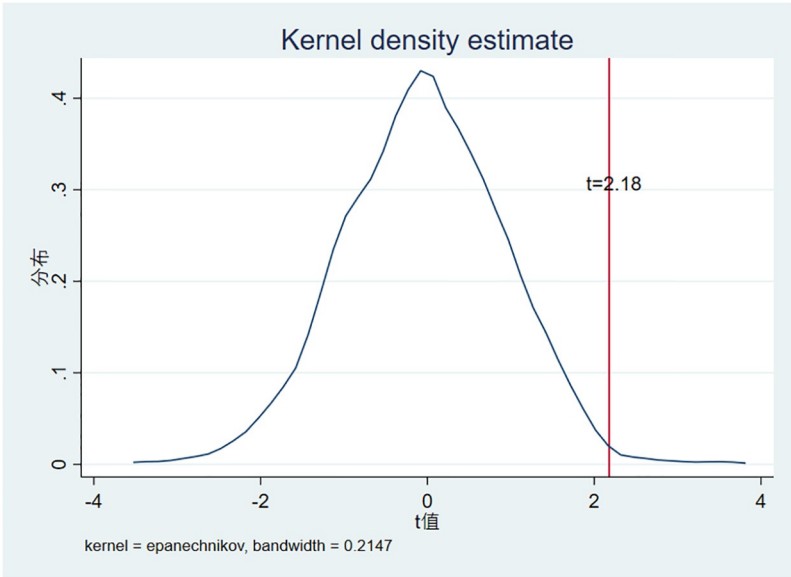

**Fig 4. The test of threshold effect.**

single threshold. The coefficient of smart cities on substantive green innovation is lower (0.014) and insignificant at low levels of marketization. When the marketization level is high, the coefficient of smart cities on substantive green innovation is higher (0.123) and significant. The regression results indicate that the level of regional marketization promotes the effect of the smart city pilot in influencing enterprises' substantive green innovation. As the level of regional marketization increases, the promotion effect improves. Hypothesis 3 is thus verified.

## 5. Discussion

This study used the DID method to explore the relationship between smart city construction and corporate green innovation. Several important findings can be drawn from the results of the study. Our study showed that the impact of smart city construction on firms' innovation behavior is bounded by the level of regional marketization. First, the findings suggest that smart city construction does promote firms to engage in innovative activities. Rather than directly influencing firms' behavior through administrative orders, changes in the environment cause firms to have to adopt corresponding behaviors, e.g., the construction of a digital economy environment prompts firms to undergo digital transformation [41].

**Table 9. Regression results of threshold model.**

|  | Lngreen_patent | P-value |
|---|---|---|
| Smart×Post (Market$_{it}$≤0.8479) | 0.014 | 0.665 |
| Smart×Post (Market$_{it}$ > 0.8479) | 0.123 | 0.000 |
| Control Variables | Yes | Yes |
| Constant | -7.984 | 0.000 |

Note: ***, **, * indicate significant at 1%, 5%, 10% confidence level.

Our findings also suggest that smart city construction affects corporate green innovation through two pathways: government innovation grants and corporate digital transformation. The findings are consistent with Guan and Hongjun (2019), who concluded that there is a positive relationship between government innovation subsidies and corporate green innovation. Smart city construction promotes the digital transformation of enterprises (Anthony et al 2021) [42]. Funding is one of the important resources that affects the innovation of enterprises; insufficient funding for enterprises affects their resource allocation on innovation, and government innovation subsidies can financially support enterprises to carry out innovative activities. The digital environment can prompt enterprises to transform their digital model from the traditional operation model, which is beneficial to the sustainable development of the city.

Finally, our results showed that the impact of smart city construction on corporate green innovation is strongly correlated with the degree of marketization. Consistent with the findings of Zeng et al. (2021). Marketization plays a very important role in firms' behavioral choices. Smart cities build a digitally smart environment, and for firms to survive and compete in the new environment, they need innovative changes. But the low degree of marketization and the lack of a greater relationship between the survival of firms and changes in the environment can constrain the innovative effects of smart city construction.

## 6. Conclusions

The results suggest that the implementation of smart cities can drive corporate green innovation. The construction of smart cities has a positive effect on the construction of an environment-friendly city. Government subsidies help promote innovative decision-making by enterprises. Meanwhile, digital transformation is also an important change made by enterprises in the process of smart city construction. Market-oriented reform plays an important role in smart cities. It can help to strengthen the chain effect brought by smart city construction by further promoting market-oriented reform and playing the moderating role of marketization in smart cities. The single marketization threshold in the threshold model is used as a criterion to distinguish between cities below and above this threshold. We focus on cities below this threshold, address the difficulties faced by these cities in the marketization process, and further promote the good effects caused by smart city construction by improving their marketization level.

## 7. Policy implications

By analyzing and demonstrating the relationship between smart city construction and corporate green innovation, we can make some important management recommendations. A digitally smart environment is an important driver of substantial green innovation for businesses. In addition, government subsidies can also drive green innovation in firms, but green innovation in firms driven by government subsidies mixes non-substantive and substantive innovation, so the use of government subsidies should be used with special care, and binding terms should be set to force firms to carry out certain substantive innovation under government subsidies.

Moreover, according to our findings, the level of marketization affects the innovation effect of smart city construction. This study argues that a high level of regional marketization can amplify the innovation effect brought by smart city construction. Conversely, a low level of regional marketization can inhibit the innovation effect of smart city construction. China has carried out market-oriented reforms for decades and has made certain achievements, but some regions still have a low level of marketization. In order to further enhance the level of marketization, the intervention of administrative forces in the market needs to be restrained,

especially in the government public sector, and the state-owned enterprise sector needs to be clearly defined to prevent state-owned enterprises from relying on administrative forces to undermine market equity. Except for strategic areas of national importance, such as finance, transportation, education, etc., it is entirely possible to regulate the allocation of resources by market forces on their own, to carry out fair competition, to enhance the vitality of enterprises, and to activate their innovation capacity.

Finally, smart city construction is the hardware foundation for the digitalization, informationalization, and intelligence of urban infrastructure. The construction of smart cities is bound to bring about the rapid development of the digital economy, which is a major opportunity to promote rapid economic growth in the future. On the government side, relevant policies, such as tax incentives, can be introduced to guide enterprises to carry out the deep integration of the digital economy and the real economy, improve the efficiency of enterprises, save energy, better serve society, and protect the environment.

## 8. Limitations and future research

It is important to note that there are some limitations to this study. This study explores the relationship between smart city construction and corporate green innovation and the role that marketization plays in it. However, this study only sets smart city construction as a dummy variable, i.e., whether to carry out smart city construction as the core explanatory variable. Smart city construction is a large and comprehensive system that includes the construction of hardware facilities as well as elements such as policy management and goal setting, and which factors play a dominant role in influencing corporate green innovation deserves further research. Future research can further explore the impact of specific aspects of smart city construction on corporate green innovation, such as smart finance, smart courts, and smart government governance. In addition, China is the location of this survey, and the study should be further extended to other economies.

## Supporting information

**S1 Data.**
(DTA)

## Author Contributions

**Conceptualization:** Zhi Zhang.

**Formal analysis:** Chengting Zheng.

**Investigation:** Longyao Lan.

**Methodology:** Zhi Zhang.

**Visualization:** Longyao Lan.

**Writing – original draft:** Zhi Zhang.

**Writing – review & editing:** Zhi Zhang, Chengting Zheng, Longyao Lan.

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
