## [Decision Letter · Decision Letter 0]

1 Nov 2022

PONE-D-22-28078Smart City Pilots, Marketization Processes, and Substantive Green Innovation: A Quasi-Natural Experiment from ChinaPLOS ONE

Dear Dr. Zhang,

Thank you for submitting your manuscript to PLOS ONE. After careful consideration, we feel that it has merit but does not fully meet PLOS ONE’s publication criteria as it currently stands. Therefore, we invite you to submit a revised version of the manuscript that addresses the points raised during the review process.

We look forward to receiving your revised manuscript.

Kind regards,

Victor Manuel Ferreira Moutinho, PhD

Academic Editor

PLOS ONE

Journal Requirements:

3. Please ensure that you refer to Figure 5 in your text as, if accepted, production will need this reference to link the reader to the figure.

4. Please upload a copy of Figure 6, to which you refer in your text on page 18. If the figure is no longer to be included as part of the submission please remove all reference to it within the text.

Reviewers' comments:

Reviewer's Responses to Questions

**Comments to the Author**

1. Is the manuscript technically sound, and do the data support the conclusions?

Reviewer #1: No

Reviewer #2: Yes

Reviewer #3: Yes

2. Has the statistical analysis been performed appropriately and rigorously? 

Reviewer #1: No

Reviewer #2: Yes

Reviewer #3: Yes

3. Have the authors made all data underlying the findings in their manuscript fully available?

Reviewer #1: No

Reviewer #2: No

Reviewer #3: Yes

4. Is the manuscript presented in an intelligible fashion and written in standard English?

Reviewer #1: No

Reviewer #2: Yes

Reviewer #3: No

5. Review Comments to the Author

Reviewer #1: I fear that this paper falls into the same trap that many similar quasi-natural experiment papers trigger: applying PSM-DID methodologies and robust techniques to data to find a pattern, without a detailed analysis of the components and dynamics of the underlying systems contributing to the academic community. The universe of time series functions is sufficiently large that something can usually be found, but this doesn´t help understand the underlying variables' interaction mechanism. This is a particularly nasty trap in correlation relationship study since very limited data about other components of its evolutionary behavior is available. Without a clear justification based on possible interaction mechanisms, which is absent, I don´t think it is valid to blindly apply techniques to investigate the task force on smart city pilots on the substantive green innovation of enterprises.

A general problem with any empirical analysis (which often gets overlooked) is finding satisfactory methods to compensate for changes in the underlying quantities of the unit of measurement. In principle, this is something that some variables mentioned in the manuscript were supposedly designed to avoid, but in practice, this has not been the case so far. I strongly advise authors to firstly construct a novel theory which is verified by the following data analyses, otherwise data results are not enough to support your conclusions. I am convinced this manuscript studied a statistical correlation, not an economic one.

I don´t find the empirical analysis compelling, or adding anything to the literature.

Overall there are a number of occasions in the paper where I think claims are being made without sufficient substantiation, and that the paper is also not robust to a suggestion of some degree of cherry picking.

Reviewer #2: 1.what's the relationship between the two or even three: Smart City Pilots, Marketization Processes, and Substantive Green Innovation. There is limited theorical analysis about this. How does Marketization Processes affect Substantive Green Innovation or vice versa.

2.How to set the hypothesis is not well founded. For example. "In reviewing the literature on smart cities, it is found that researchers generally agree that smart cities create a system of innovation". It's suggested to gather theorical model or analysis to support the hypothesis.

The empirical analysis is only part of verifying the theory not a substitution of theory.

3.the Moderating effect test is not sufficient without a theorical analysis.

4.the endogenous problem must be addressed.

Reviewer #3: The study found that smart city pilots drove substantial green innovation in businesses. Marketization process has moderating effect on the impact of smart city pilots on substantive green innovation of enterprises. Through further analysis, it is found that the marketization process has threshold effect in smart city pilots influencing the green substantive innovation of enterprises, and the effect of smart city driver influencing the substantive green innovation of enterprises increases significantly when regional marketization process reached a certain level. Lastly, based on the empirical results, relevant suggestions for optimizing smart city construction are proposed to lay the foundation for further exerting the good effect of smart cities. The topic is interesting for journal readers. The suggestion is to improve the discussion of the results in terms of policy implications. Moreover, the text should be English proofread because some sentences are not clear.

6. PLOS authors have the option to publish the peer review history of their article (what does this mean?). If published, this will include your full peer review and any attached files.

Reviewer #1: No

Reviewer #2: No

Reviewer #3: No

---

## [Author Response · Author response to Decision Letter 0]

10 Dec 2022

Dear editor and reviewers,

Thank you for offering us an opportunity to improve the quality of our submitted manuscript (PONE-D-22-28078). We appreciated very much the reviewers constructive and insightful comments. In this revision, we have addressed all of these comments. We hope the revised manuscript has now met the publication standard of PLOS ONE.

Reviewer #1

Comment 1: I fear that this paper falls into the same trap that many similar quasi-natural experiment papers trigger: applying PSM-DID methodologies and robust techniques to data to find a pattern, without a detailed analysis of the components and dynamics of the underlying systems contributing to the academic community. The universe of time series functions is sufficiently large that something can usually be found, but this doesn´t help understand the underlying variables' interaction mechanism. This is a particularly nasty trap in correlation relationship study since very limited data about other components of its evolutionary behavior is available. Without a clear justification based on possible interaction mechanisms, which is absent, I don´t think it is valid to blindly apply techniques to investigate the task force on smart city pilots on the substantive green innovation of enterprises. A general problem with any empirical analysis (which often gets overlooked) is finding satisfactory methods to compensate for changes in the underlying quantities of the unit of measurement. In principle, this is something that some variables mentioned in the manuscript were supposedly designed to avoid, but in practice, this has not been the case so far. I strongly advise authors to firstly construct a novel theory which is verified by the following data analyses, otherwise data results are not enough to support your conclusions. I am convinced this manuscript studied a statistical correlation, not an economic one. I don´t find the empirical analysis compelling, or adding anything to the literature. Overall, there are a number of occasions in the paper where I think claims are being made without sufficient substantiation, and that the paper is also not robust to a suggestion of some degree of cherry picking.

Response: Thanks for the suggestions. The study we conducted was based primarily on the theory of government behavior and firm innovation, because China's smart city pilot is a typically integrated government action, there are strict requirements for companies in terms of digitalization and environmental protection. We have added the theoretical framework of government intervention behavior and firm innovation to the section 2.1. In the section 4.3, we have added potential mechanisms for smart city pilots to influence firm innovation: government subsidies to firms and digital transformation of firms.

Reviewer #2: 

1.what's the relationship between the two or even three: Smart City Pilots, Marketization Processes, and Substantive Green Innovation. There is limited theorical analysis about this. How does Marketization Processes affect Substantive Green Innovation or vice versa.

Response: Thanks for the suggestions. The study we conducted was based primarily on the theory of government behavior and firm innovation. Smart city construction as a governmental action can affect firm innovation. In section 2.1 we have added the theoretical framework of government intervention behavior and firm innovation. In different cities, the degree of marketization varies, leading to different effects of smart city pilots on firm innovation. In Section 2.3, we analyze the theoretical basis of marketization as a moderator. We verify, in section 4.4 of the paper, the moderating effect of marketization through a moderating effect model. In fact, we want to explain that the government as the "visible hand" and the "invisible hand" of the market mechanism to cooperate with each other in order to build a conducive environment for the healthy development of enterprises.

2.How to set the hypothesis is not well founded. For example. "In reviewing the literature on smart cities, it is found that researchers generally agree that smart cities create a system of innovation". It's suggested to gather theorical model or analysis to support the hypothesis. The empirical analysis is only part of verifying the theory not a substitution of theory.

Response: Thanks for the suggestions. We have added the theoretical framework of government intervention behavior and firm innovation to the section 2.1. In the section 4.3, we have added potential mechanisms for smart city pilots to influence firm innovation: government subsidies to firms and digital transformation of firms.

3.the Moderating effect test is not sufficient without a theorical analysis.

Response: Thanks for the suggestions. We have added a theoretical analysis of the moderating effect to Section 2.3.

4.the endogenous problem must be addressed.

Response: Thanks for the suggestions. Regarding the endogeneity problem, we used a combination of PSM and DID. The PSM method can match each treatment group sample to a specific control group sample, making the quasi-natural experiment approximately random, thus well solving the endogeneity problem of selective samples.

Reviewer #3: The study found that smart city pilots drove substantial green innovation in businesses. Marketization process has moderating effect on the impact of smart city pilots on substantive green innovation of enterprises. Through further analysis, it is found that the marketization process has threshold effect in smart city pilots influencing the green substantive innovation of enterprises, and the effect of smart city driver influencing the substantive green innovation of enterprises increases significantly when regional marketization process reached a certain level. Lastly, based on the empirical results, relevant suggestions for optimizing smart city construction are proposed to lay the foundation for further exerting the good effect of smart cities. The topic is interesting for journal readers. The suggestion is to improve the discussion of the results in terms of policy implications. Moreover, the text should be English proofread because some sentences are not clear.

Response: Thanks for the suggestions. We've added a part for government-specific recommendations in section 6.

---

## [Decision Letter · Decision Letter 1]

20 Mar 2023

PONE-D-22-28078R1Smart City Pilots, Marketization Processes, and Substantive Green Innovation: A Quasi-Natural Experiment from ChinaPLOS ONE

Dear Dr. Zhang,

Thank you for submitting your manuscript to PLOS ONE. After careful consideration, we feel that it has merit but does not fully meet PLOS ONE’s publication criteria as it currently stands. Therefore, we invite you to submit a revised version of the manuscript that addresses the points raised during the review process.

We look forward to receiving your revised manuscript.

Kind regards,

Xingwei Li, Ph.D.

Academic Editor

PLOS ONE

Reviewers' comments:

Reviewer's Responses to Questions

**Comments to the Author**

1. If the authors have adequately addressed your comments raised in a previous round of review and you feel that this manuscript is now acceptable for publication, you may indicate that here to bypass the “Comments to the Author” section, enter your conflict of interest statement in the “Confidential to Editor” section, and submit your "Accept" recommendation.

Reviewer #1: All comments have been addressed

Reviewer #3: All comments have been addressed

Reviewer #4: (No Response)

Reviewer #5: (No Response)

Reviewer #6: (No Response)

2. Is the manuscript technically sound, and do the data support the conclusions?

Reviewer #1: Yes

Reviewer #3: Yes

Reviewer #4: Partly

Reviewer #5: Partly

Reviewer #6: Partly

3. Has the statistical analysis been performed appropriately and rigorously? 

Reviewer #1: Yes

Reviewer #3: Yes

Reviewer #4: (No Response)

Reviewer #5: No

Reviewer #6: Yes

4. Have the authors made all data underlying the findings in their manuscript fully available?

Reviewer #1: Yes

Reviewer #3: Yes

Reviewer #4: (No Response)

Reviewer #5: Yes

Reviewer #6: Yes

5. Is the manuscript presented in an intelligible fashion and written in standard English?

Reviewer #1: Yes

Reviewer #3: Yes

Reviewer #4: (No Response)

Reviewer #5: No

Reviewer #6: Yes

6. Review Comments to the Author

Reviewer #1: The study found that smart city pilots drove substantial green innovation in businesses. Marketization process has moderating effect on the impact of smart city pilots on substantive green innovation of enterprises.

All comments are replied.

The current version is suitable for publication.

Reviewer #3: The paper has been improved according to the reviewers' comments. Now the manuscript can be accepted for publication.

Reviewer #4: Thank you very much for the opportunity to review the manuscript entitled “Smart City Pilots, Marketization Processes, and Substantive Green Innovation: A Quasi-Natural Experiment from China” (PONE-D-22-28078R1). This research applied a quasi-natural experiment of the PSM-DID method to explore the impact of smart city pilots on the substantive green innovation of enterprises. It is a very interesting topic and offers insightful findings for optimizing smart city construction. The paper could make a potential contribution, there are, however, a few questions to be clarified:

1. Although the introduction does a good job as explaining the importance of smart city construction, it does not clearly pose the research question, why did the paper aim to find out the intrinsic relationship between the construction of smart city and substantive green innovation of enterprises, and why did the paper take marketization into consideration? Besides, the three contributions need more work to be highlighted and clarified based on specific gaps, and suggesting that contributions can be elaborated in the section of Introduction.

2. Although the revised paper has added the theoretical framework in the section 2.1, it still failed to well examine the possible relationship between smart cities and green innovation. Specifically, I agree with the idea that smart city pilot is a kind of government behaviors, while not each government behavior will affect green innovation, thus suggesting that its theoretical logic should be explained much more in detail in the sections of Theoretical framework and Hypothesis development.

3. Robustness tests were mainly according to the main effect of smart cities on green innovation, so it is better that the section 5 could be removed to behind of section 4.2. Additionally, besides PSM, other methods could be suggested to test endogeneity comprehensively.

4. The revised version has added the underlying plausible mechanism for smart city pilots to influence firm innovation: government subsidies to firms and digital transformation of firms in the section 4.3, while there is no section 2.4 in this paper as mentioned on page 14, so the explanations of mediating variables or potential mechanisms were missing.

5.This paper lacks a deep discussion to engage the literature, suggesting you could compare your study with the relevant literature in details, and show why your findings matter theoretically. Additionally, some limitations or future directions could be proposed.

Reviewer #5: (No Response)

Reviewer #6: The manuscript is subject to the following major revisions:

Abstract: The abstract should be written in the following context: Background, objective(s), methods, results, conclusions, policy recommendations.

Introduction: The introduction lacks study background. What is the novelty? How is your study different from other studies? Please explain in detail.

Many important studies related to sustainable development have been ignored. For instance, consult the following studies and improve your study:

https://doi.org/10.1007/s11356-023-25662-w

https://doi.org/10.1080/1331677X.2022.2159849

https://doi.org/10.1007/s11356-022-24899-1

https://doi.org/10.1007/s11356-022-24387-6

https://doi.org/10.1007/s11356-022-24286-w

How you can compare your results with other studies of same geographical regions. It is pivotal to tie your results with other relevant literature.

Discussion: The study discussion is very generic and do not stem from study findings. It is suggested to rearrange discussion based on study findings.

Conclusions: Conclusions are very weak and miss several important dimensions. To strengthen the contents and quality of the study, conclusions must be revised for more clarity and for the ease of normal readers.

Policy recommendations: Specific policy recommendations should be put forward according to the target sample. General policies are of no use in scholarly articles.

Study limitations should be provided along with future research directions for prospective scholars interested in the similar works.

The authors have used several old references to support their arguments. We are in 2023 and you are using such old references. In order to nurture the importance of study, references should be updated using recent and relevant studies.

There is an intermingle of capital and small letters. Please avoid this practice in scientific writing.

Finally, the manuscript can be benefited if the authors thoroughly proofread it in terms of English language mistakes and syntax structure.

7. PLOS authors have the option to publish the peer review history of their article (what does this mean?). If published, this will include your full peer review and any attached files.

Reviewer #1: No

Reviewer #3: No

Reviewer #4: No

Reviewer #5: No

Reviewer #6: No

---

## [Author Response · Author response to Decision Letter 1]

17 Apr 2023

Dear editor and reviewers,

Thank you for offering us an opportunity to improve the quality of our submitted manuscript (PONE-D-22-28078). We appreciated very much the reviewers constructive and insightful comments. In this revision, we have addressed all these comments. We hope the revised manuscript has now met the publication standard of PLOS ONE.

Reviewer #4: 

1. Although the introduction does a good job as explaining the importance of smart city construction, it does not clearly pose the research question, why did the paper aim to find out the intrinsic relationship between the construction of smart city and substantive green innovation of enterprises, and why did the paper take marketization into consideration? Besides, the three contributions need more work to be highlighted and clarified based on specific gaps, and suggesting that contributions can be elaborated in the section of Introduction.

Response: Thanks for the suggestions. We have added lines 29–74 in the revised version, detailing the goal of the article: to explore the relationship between smart city construction and enterprise green innovation. Lines 68–74 explain why marketization is being added to examine the connection between smart city construction and corporate innovation. Lines 75–86 provides a detailed explanation of the gaps and the contributions of this article.

2. Although the revised paper has added the theoretical framework in the section 2.1, it still failed to well examine the possible relationship between smart cities and green innovation. Specifically, I agree with the idea that smart city pilot is a kind of government behaviors, while not each government behavior will affect green innovation, thus suggesting that its theoretical logic should be explained much more in detail in the sections of Theoretical framework and Hypothesis development.

Response: Thanks for the suggestions. In the revised version, we explained much more in detail in Section 2 about theoretical logic and hypothesis development. We explained in more detail on lines 92–144 how the government's smart city construction affects the innovative behavior of enterprises.

3. Robustness tests were mainly according to the main effect of smart cities on green innovation, so it is better that the section 5 could be removed to behind of section 4.2. Additionally, besides PSM, other methods could be suggested to test endogeneity comprehensively.

Response: Thanks for the suggestions. In the revised version, the section on robustness testing was moved behind Section 4.2, according to your suggestions, and the IV method for the robustness test was added. Therefore, we use the PSM method in combination with the DID and IV methods to comprehensively solve endogeneity problems. The detailed steps and results are shown on lines 207–257.

4. The revised version has added the underlying plausible mechanism for smart city pilots to influence firm innovation: government subsidies to firms and digital transformation of firms in the section 4.3, while there is no section 2.4 in this paper as mentioned on page 14, so the explanations of mediating variables or potential mechanisms were missing.

Response: Thanks for the suggestions. This is due to a change in the structure of the paper, resulting in a change in chapter position, which was our mistake. Regarding the explanations of mediating variables or potential mechanisms on lines 107–140.

5.This paper lacks a deep discussion to engage the literature, suggesting you could compare your study with the relevant literature in details, and show why your findings matter theoretically. Additionally, some limitations or future directions could be proposed.

Response: Thanks for the suggestions. In the new revised version, discussion is displayed on lines 331–351.We have added the "Limitations and future research" section on lines 376–384.

Reviewer #5: 

1. the writing structure of the Chinese paper. The authors should read more English papers and understand the English paper writing structure instead of directly translating Chinese papers into English. As an example, the article lacks a follow-up research arrangement in the introduction section.

Response: Thanks for the suggestions. In the new revised version, we have redesigned according to the writing conventions of English papers. Specifically, first we added a follow-up research arrangement in the introduction section. Secondly, the specific chapters of the paper include the following parts: Section 1 is an introduction; Section 2 is a literature review and hypothesis development; and Section 3 presents the research methodology, data, and variable definitions; Section 4 contains the empirical results; Section 5 is the discussion; Section 6 outlines the policy implications; and Section 7 is about the limitations and future research.

2. the references are too old. Most of the papers are before 2019, and many are even 2009. hopefully, the authors can recognize the importance of the literature review. In addition, I hope you can compare with the existing literature in the literature review to highlight the strengths of your research.

Response: Thanks for the suggestions. In the new revised version, we have redesigned the literature review section. We have deleted some old literature and added important new literature, as reflected in Sections 1 and 2 of the paper.

3. the authors should follow the structure of research objectives-methods-results-conclusion in the writing of the abstract.

Response: Thanks for the suggestions. In the new revised version, we have considered your and other reviewers' opinions and rewritten the abstract section according to Background, Objective (s), Methods, Results, Conclusions, Policy Recommendations. Specifically, on lines 10-22.

4. The introduction is really poorly written. After reading the introduction I didn't understand the background of your study and the progress of the current study. The introduction is a very important part of the article. However, the author has neglected to do so. In the introduction, the author should give an overview of the background, significance and development of the study. Point out the important advances in current research and the gaps in current research. Based on this, introduce the purpose of their research. Finally highlight their research contribution and summarize the overall paper's research.

Response: Thanks for the suggestions. In the new revised version, we have rewritten the introduction. Specifically, we give an overview of the background, significance, and development of the study, point out the important advantages of current research and the gaps in current research, highlight research contributions, and summarize the overall paper's research. Specifically reflected in lines 26-90.

5. Hypotheses 2 and 3 are formulated too arbitrarily and without any theoretical support at all.

Response: Thanks for the suggestions. In the new revised version, we have rewritten the hypothesis section. Specifically, we provided a more detailed explanation of the theoretical logic behind the hypothesis. Specifically reflected in lines 92–140.

6. The author's methodological design is too arbitrary and I hope it can be redesigned.

Response: Thanks for the suggestions. In the new revised version, we use the PSM method in combination with the DID and IV methods to comprehensively verify the conclusions of the paper.

7. I did not find fig1 and fig2, however, the authors cited two figures.

Response: Thanks for the suggestions. In the new revised version, figs. 1 and 2 are behind page 28. It should be noted that the PDF file formed in the editing system contains two versions of the paper: one is the normal revised manuscript and the other is the revised manuscript with tracked changes. The former has figures, and the latter does not.

8. the authors' conclusion is poorly written. the conclusion section briefly summarizes the study of the paper and then points out the shortcomings of the study.

Response: Thanks for the suggestions. In the new revised version, we have redesigned the structure of the paper, and the last three parts of the paper include discussion, policy implications, limitations, and future research, as detailed in lines 330–384.

9. The authors' study lacks a discussion section.

Response: Thanks for the suggestions. In the new revised version, we have redesigned the structure of the paper, and the last three parts of the paper include discussion, policy implications, limitations, and future research, as detailed in lines 330–384.

Reviewer #6: 

1. Abstract: The abstract should be written in the following context: Background, objective(s), methods, results, conclusions, policy recommendations.

Thanks for the suggestions. In the new revised version, we have considered your and other reviewers' opinions and rewritten the abstract section according to Background, Objective (s), Methods, Results, Conclusions, Policy Recommendations. Specifically, on lines 10-22.

2.Introduction: The introduction lacks study background. What is the novelty? 

Thanks for the suggestions. In the new revised version, we have rewritten the introduction. Specifically, we give an overview of the background, significance, and development of the study, point out the important advantages of current research and the gaps in current research, highlight research contributions, and summarize the overall paper's research. Specifically reflected in lines 26-90.

3.How is your study different from other studies? Please explain in detail.

Thanks for the suggestions. Firstly, we examine the impact of smart city construction on microenterprises; previous research has mostly focused on the macro level. Secondly, we have added a very important element of marketization to the construction of smart cities and enterprise innovation behavior, as detailed in lines 56–86.

4.Many important studies related to sustainable development have been ignored. For instance, consult the following studies and improve your study:

https://doi.org/10.1007/s11356-023-25662-w

https://doi.org/10.1080/1331677X.2022.2159849

https://doi.org/10.1007/s11356-022-24899-1

https://doi.org/10.1007/s11356-022-24387-6

https://doi.org/10.1007/s11356-022-24286-w

Response: Thanks for the suggestions. In the new revised version, we have redesigned the literature review section. We have deleted some old literature and added important new literature, as reflected in Sections 1 and 2 of the paper.

5.How you can compare your results with other studies of same geographical regions. It is pivotal to tie your results with other relevant literature.

Response: Thanks for the suggestions. In the section of discussion, we tied our results with other relevant literature, as shown in lines 331-351.

6.Discussion: The study discussion is very generic and do not stem from study findings. It is suggested to rearrange discussion based on study findings.

Response: Thanks for the suggestions. In the new revised version, we have rearranged the discussion based on study findings as detailed in lines 331–351.

7.Conclusions: Conclusions are very weak and miss several important dimensions. To strengthen the contents and quality of the study, conclusions must be revised for more clarity and for the ease of normal readers.

Response: Thanks for the suggestions. In the new revised version, we have redesigned the structure of the paper, and the last three parts of the paper include discussion, policy implications, limitations, and future research, as detailed in lines 330–384. It is more clarity and for the ease of normal readers.

8.Policy recommendations: Specific policy recommendations should be put forward according to the target sample. General policies are of no use in scholarly articles.

Response: Thanks for the suggestions. In the new revised version, we have redesigned the section on policy implications, as detailed in lines 353–374. It is more specific and detailed.

9.Study limitations should be provided along with future research directions for prospective scholars interested in the similar works.

Response: Thanks for the suggestions. In the new revised version, we have added the section on limitations and future research, as detailed in lines 375–384.

10.The authors have used several old references to support their arguments. We are in 2023 and you are using such old references. In order to nurture the importance of study, references should be updated using recent and relevant studies.

Response: Thanks for the suggestions. In the new revised version, we have redesigned the literature review section. We have deleted some old literature and added important new literature, as reflected in Sections 1 and 2 of the paper.

11.There is an intermingle of capital and small letters. Please avoid this practice in scientific writing.

Response: Thanks for the suggestions. In the new revised version, we have checked and corrected the grammar and letter capitalization issues throughout the entire paper.

12. Finally, the manuscript can be benefited if the authors thoroughly proofread it in terms of English language mistakes and syntax structure.

Response: Thanks for the suggestions. In the new revised version, we have thoroughly proofread it in terms of English language mistakes and syntax structure.

---

## [Decision Letter · Decision Letter 2]

15 May 2023

PONE-D-22-28078R2Smart City Pilots, Marketization Processes, and Substantive Green Innovation: A Quasi-Natural Experiment from ChinaPLOS ONE

Dear Dr. Zhang,

Thank you for submitting your manuscript to PLOS ONE. After careful consideration, we feel that it has merit but does not fully meet PLOS ONE’s publication criteria as it currently stands. Therefore, we invite you to submit a revised version of the manuscript that addresses the points raised during the review process.

We look forward to receiving your revised manuscript.

Kind regards,

Xingwei Li, Ph.D.

Academic Editor

PLOS ONE

Journal Requirements:

Additional Editor Comments:

The section structure of this manuscript does not make sense. Please redesign the section structure, a conclusion section should be located after the discussion section. Perhaps you should reorganize the layout of the section structure for Conclusions, Policy implications and Limitations and future research.

Reviewers' comments:

Reviewer's Responses to Questions

**Comments to the Author**

1. If the authors have adequately addressed your comments raised in a previous round of review and you feel that this manuscript is now acceptable for publication, you may indicate that here to bypass the “Comments to the Author” section, enter your conflict of interest statement in the “Confidential to Editor” section, and submit your "Accept" recommendation.

Reviewer #4: (No Response)

Reviewer #5: All comments have been addressed

Reviewer #6: All comments have been addressed

2. Is the manuscript technically sound, and do the data support the conclusions?

Reviewer #4: (No Response)

Reviewer #5: Yes

Reviewer #6: Yes

3. Has the statistical analysis been performed appropriately and rigorously? 

Reviewer #4: (No Response)

Reviewer #5: Yes

Reviewer #6: Yes

4. Have the authors made all data underlying the findings in their manuscript fully available?

Reviewer #4: (No Response)

Reviewer #5: Yes

Reviewer #6: Yes

5. Is the manuscript presented in an intelligible fashion and written in standard English?

Reviewer #4: (No Response)

Reviewer #5: Yes

Reviewer #6: Yes

6. Review Comments to the Author

Reviewer #4: This review concerns the revised manuscript titled “Smart City Pilots, Marketization Processes, and Substantive Green Innovation: A Quasi-Natural Experiment from China” (PONE-D-22-28078R2). The authors have made more efforts to improve the writing quality and revise the concerns mentioned by the reviewers. In this revised manuscript, I feel that it can be accepted for publication in this journal.

Reviewer #5: (No Response)

Reviewer #6: (No Response)

7. PLOS authors have the option to publish the peer review history of their article (what does this mean?). If published, this will include your full peer review and any attached files.

Reviewer #4: No

Reviewer #5: No

Reviewer #6: No

---

## [Author Response · Author response to Decision Letter 2]

17 May 2023

Dear editor and reviewers,

Thank you for offering us an opportunity to improve the quality of our submitted manuscript (PONE-D-22-28078). We appreciated very much the reviewers constructive and insightful comments. In this revision, we have addressed all these comments. We hope the revised manuscript has now met the publication standard of PLOS ONE.

Journal Requirements:

Thanks for the suggestions. We checked all references one by one and corrected them as follows:

9. Han Y. Smart Growth, Smart City. In: 2017 International Conference on Machinery, Electronics and Control Simulation (mecs 2017). Paris: Atlantis Press; 2017. p. 580–3. 

22. Shelton T, Zook M, Wiig A. The ‘actually existing smart city.’ Cambridge Journal of Regions. 2015; 8:13-25. https://doi.org/10.1093/cjres/rsu026

28. Nam T, Pardo TA. Conceptualizing smart city with dimensions of technology, people, and institutions. In: the 12th Annual International Digital Government Research Conference on Digital Government Innovation in Challenging Times. Maryland: ACM Press; 2011. p. 282-91. https://doi.org/10.1145/2037556.2037602

32. Sun B, Ruan A, Peng B, Liu S. Pay disparities within top management teams, marketization and firms’ innovation: evidence from China. Journal of the Asia Pacific Economy. 2022;715–35. https://doi.org/10.1080/13547860.2020.1865248

Additional Editor Comments:

The section structure of this manuscript does not make sense. Please redesign the section structure, a conclusion section should be located after the discussion section. Perhaps you should reorganize the layout of the section structure for Conclusions, Policy implications and Limitations and future research.

Response: Thanks for the suggestions. The conclusion section has been located after the discussion section in the revised version. Meanwhile, we reintroduced the structure of the article in the introduction section. Specifically reflected in lines 87–90 and 352-361. 

Reviewer #1: All comments have been addressed

Reviewer #3: All comments have been addressed

Reviewer #4: This review concerns the revised manuscript titled “Smart City Pilots, Marketization Processes, and Substantive Green Innovation: A Quasi-Natural Experiment from China” (PONE-D-22-28078R2). The authors have made more efforts to improve the writing quality and revise the concerns mentioned by the reviewers. In this revised manuscript, I feel that it can be accepted for publication in this journal.

Reviewer #5: All comments have been addressed

Reviewer #6: All comments have been addressed

We sincerely thank the reviewers and editor for their valuable comments, which are of constructive significance to the improvement of this paper.

---

## [Editor Report · Decision Letter 3]

19 May 2023

Smart City Pilots, Marketization Processes, and Substantive Green Innovation: A Quasi-Natural Experiment from China

PONE-D-22-28078R3

Dear Dr. Zhang,

We’re pleased to inform you that your manuscript has been judged scientifically suitable for publication and will be formally accepted for publication once it meets all outstanding technical requirements.

Kind regards,

Xingwei Li, Ph.D.

Academic Editor

PLOS ONE

Additional Editor Comments (optional):

The authors have carefully revised the manuscript in accordance with the comments, and the current version is acceptable.
---

## [Editor Report · Acceptance letter]

24 May 2023

PONE-D-22-28078R3 

Smart City Pilots, Marketization Processes, and Substantive Green Innovation: A Quasi-Natural Experiment from China 

Dear Dr. Zhang:

I'm pleased to inform you that your manuscript has been deemed suitable for publication in PLOS ONE. Congratulations! Your manuscript is now with our production department. 

Kind regards, 

on behalf of

Prof. Dr. Xingwei Li 

Academic Editor

PLOS ONE